# Two New Nematode Species, *Desmolaimus magnus* sp. nov. (Monhysterida, Linhomoeidae) and *Metadesmolaimus robustus* sp. nov. (Monhysterida, Xyalidae), from the Yellow Sea, China with Phylogenetic Analyses within Linhomoeidae and Xyalidae†

**Wen Guo, Zhiyu Meng and Chunming Wang \*** 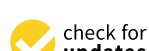

College of Life Sciences, Liaocheng University, Liaocheng 252059, China; lcuguowen@outlook.com (W.G.); mengzhiyulcu@outlook.com (Z.M.)

\* Correspondence: wangchunming@lcu.edu.cn

† lsid:zoobank.org:act:D5D54F33-2679-49D5-A3A4-F38E499BA3ED;
lsid:zoobank.org:act:901DC71D-C907-441A-A4B7-4E146D699F27

**Abstract:** Two new species are described from the Rizhao coast along the Yellow Sea. *Desmolaimus magnus* sp. nov. is characterized by its relatively large body size, faintly striated cuticle, four long cephalic setae, a wide buccal cavity with cuticularized transverse rings, an amphidial fovea at the junction of the buccal cavity and pharynx, a pharynx without a bulb, curved spicules, a gubernaculum with a dorso–caudal apophysis, and an elongated conical tail. *Metadesmolaimus robustus* sp. nov. is characterized by its relatively large body size, coarsely striated cuticle, spacious buccal cavity with cuticularized rings, six inner labial sensilla setiform, six outer labial setae, and four long cephalic setae. The pharynx is notably muscularized, and the spicules are straight and jointed with a slightly swollen proximal end and a hooked distal end. The gubernaculum is brownish and comma–shaped, and the tail is conico–cylindrical with terminal setae. Phylogenetic analyses using maximum–likelihood and Bayesin inference, based on small subunit and D2–D3 fragment of large subunit rDNA sequences, place *Desmolaimus magnus* sp. nov. within the framework of the family Linhomoeidae and *Metadesmolaimus robustus* sp. nov. within Xyalidae. Topology trees at the family level show genera *Desmolaimus* and *Metadesmolaimus* as paraphyletic groups, the genus *Terschellingia* as a monophylectic group, the genus *Theristus* as a monophylectic group based on SSU sequence, the genus *Daptonema* as a monophylectic group based on LSU sequence.

**Keywords:** free-living marine nematodes; phylogenetic analysis; SSU; LSU; taxonomy

## 1. Introduction

Nematodes are the most abundant and diverse group among meiofauna animals and play important roles as various trophic guilds. The estimated number of nematode species ranges from half a million to ten million, and only 1% to 5% of these species have been identified [1]. Phylogenetic relationships among nematodes are still not well understood due to the scarcity of available molecular sequences deposited in GenBank. Linhomoeidae Filipjev, 1922, and Xyalidae Chitwood, 1951, are the two most complex families in the order Monhysterida. Genera among them are morphological similar, and genus relationships between *Metadesmolaimus* Schuurmans Stekhoven, 1935 and *Daptonema* Cobb, 1920, as well as between *Metadesmolaimus* and *Theristus* Bastian, 1865, have been a source of confusion for a long time [2–4].

The genus *Desmolaimus* was established in 1880 by de Man, based on the following morphological characteristics of the type species, *D. zeelandicus* de Man, 1880: non-striated cuticle, an undistinguished head, circular amphids, a small, cup-shaped buccal cavity

with cuticularized ridges, a pharynx with a posterior bulb, and a symmetrical, elongated female reproductive system [5]. Kreis revised the genus characteristics based on head sensilla, amphidial fovea shape, cardia, and tail shape, and described *D. longicaudatus* Kreis, 1929 [6]. After Kreis, *D. zosterae* Allgén, 1933, *D. calvus* Gerlach, 1956, *D. bibulbosus* Allgén, 1959, *D. conicaudatus* Allgén, 1959, *D. greenpatchi* Allgén, 1959, *D. macrocirculus* Allgén, 1959, and *D. propinquus* Allgén, 1959 were described. Schneider transferred *D. brachystoma* (Hofmänner & Menzel, 1914) Micoletzky, 1925 and *D. thienemanni* Micoletzky, 1922 to the genus *Hofmaenneria* Gerlach & Meyl, 1957 [7]. Gerlach discussed the position of *Desmolaimus* within the family Linhomoeidae Filipjev, 1922 based on cephalic setae, cervical setae, amphidial fovea, and buccal cavity. He revised the genus's character, mainly based on head sensilla and subcephalic setae. He described *D. brasiliensis* Gerlach, 1963, synonymized *D. demani* Schulz, 1932 and *D. fennicus* (Schneider, 1926) Gerlach, 1953 as *D. zeelandicus*, and considered *D. mirabilis* Allgén, 1935, *D. tristis* Allgén, 1935, *D. bibulbosus*, and *D. greenpatchi* as species inquirenda [8]. After Gerlach, *D. bulbulus* Lorenzen, 1969, *D. courti* Leduc & Gwyther, 2008 and *D. minor* Gagarin, 2019 were described. The genus relationship between *Desmolaimus* and *Metadesmolaimus* Schuurmans Stekhoven, 1935 was discussed by Leduc & Gwyther, and transverse cuticularized rings in the buccal cavity were considered a reliable distinguishing character from *Metadesmolaimus* [9]. Gagarin considered *D. conicaudatus*, *D. macrocirculus*, and *D. propinquus* as species inquirenda because they were not fully described, especially description the buccal cavity structure absent, and the detailed illustrations not provided [10]. As of now, eight species have been considered as valid.

The genus *Metadesmolaimus* was first erected by Schuurmans Stekhoven in 1935 based on *M. labiosetosus* (Schuurmans Stekhoven, 1935) with the genus character of labial sensilla in three circles, a circular amphidial fovea, a buccal cavity similar to *Sphaerolaimus* Bastian, 1865 and *Desmolaimus*, and a tail elongated and conical [11]. But *M. labiosetosus* was based on a single juvenile specimen and considered as species inquirenda. Later, Gerlach described *M. aversivulva* and differentiated the genus from *Theristus* mainly based on the buccal cavity structure [12]. Lorenzen described *M. aduncus* Lorenzen, 1971, *M. pandus* Lorenzen, 1971, *M. heteroclitus* Lorenzen, 1971, *M. varians* Lorenzen, 1971 and transferred *Paramonhystera canicula* (Wieser & Hopper, 1967), *Theristus hamatus* (Gerlach 1956) to genus *Metadesmolaimus* [2]. Gerlach & Riemann transferred *Paramonhystera canicula* to *Metadesmolaimus* [13]. Lorenzen systematically reviewed *Metadesmolaimus* considering cuticle, cephalic setae, gonads, and tail shape. Lorenzen also redescribed and transferred *Daptonema gelana* (Warwick & Platt, 1973) to *Metadesmolaimus* based on the presence of two bristle-shaped structures [14]. Later, *M. gaelicus* Platt, 1983, *M. similis* Tchesunov, 1990, *M. psammophilus* Tchesunov, 1990, *M. communis* Gagarin, 2013, and *M. elegans* Gagarin, 2013 were described. Tchesunov & Miljutin discussed the relationship between *Metadesmolaimus* and *Daptonema* based on the structure of the copulatory apparatus and noted that morphological characters between these two genera are overlapping and loosely defined [3]. Guo et al. recently described *M. zhanggi* Guo Chen & Liu, 2016 and further discussed the genus characteristics, including the buccal cavity with an extended cylindrical anterior section, the existence of ventrolateral soft setiform structure between the labial and cephalic setae, and the brownish cuticle colour [15]. As of now, 15 species have been considered as valid.

During our study of marine nematode diversity in the Rizhao coast, Yellow Sea, China, two new species, *Desmolaimus magnus* sp. nov. and *Metadesmolaimus robustus* sp. nov., are described, and sequences of SSU and the D2–D3 fragment of LSU rDNA are acquired. Phylogenetic analyses within the family Linhomoeidae and Xyalidae are conducted to clarify their classification at the generic level.

## 2. Materials and Methods

### 2.1. Sample Collection and Nematode Identification

In May 2022, undisturbed sediment samples were collected using a syringe with a 2.6 cm inner diameter to a depth of 10 cm from the Rizhao coast in the Yellow Sea, China.

Samples for morphological analysis were fixed in a 10% formalin solution in seawater and samples for molecular analysis were preserved in 95% ethanol. In the laboratory, the formalin-fixed samples underwent a process involving washing through two sieves with mesh sizes of 500 μm and 45 μm, effectively separating meiofauna from macrofauna (organisms larger than 500 μm). The separated meiofauna were carefully transferred to a grid–lined Petri dish and sorted using a stereoscopic microscope. Nematodes were then transferred into a mixture of ethanol (50%) and glycerin (in a 1:9 volume ratio) allowing the ethanol to slowly evaporate [16]. The nematodes were subsequently mounted in glycerin on permanent slides. Descriptions were made using an Axiscope–5 differential interference contrast microscope (Zeiss, Jena, Germany) with a camera of Axiocam 208 color, line drawings were created with ZEN 3.1 Labscope software through an iPad (Apple, Fresno, CA, USA), and measurements and photographs were taken with the aid of ZEN 3.1 software (Zeiss). The type specimens were deposited at the Institute of Oceanology, Chinese Academy of Sciences, Qingdao.

*2.2. DNA Extraction, PCR Amplification, and Phylogenetic Analysis*

Samples used for molecular analysis were washed and separated as formalin-fixed samples. One male of *D. magnus* sp. nov. was sorted and confirmed primarily based on buccal cavity, amphidial fovea, spicules, gubernaculum, and tail shape on the temporary slides. Five males of *M. robustus* sp. nov. were sorted and confirmed primarily based on the buccal cavity, spicules, gubernaculum, and tail shape on the temporary slides.

Genomic DNA of two species was extracted with the DNeasy Blood & Tissue kit (Qiagen, Hilden, Germany) and used as amplification templates for nearly full length SSU rDNA sequence with primers G18S4F (5′-GCT TGT CTC AAA GAT TAA GCC-3′)/18PR (5′-TGA TCC WMC RGC AGG TTC AC-3′) [17], and for the D2–D3 fragments of LSU rDNA sequence with primers D2A (5′-ACA AGT ACC GTG AGG GAA AGT TG-3′)/D3B (5′-TCG GAA GGA ACC AGC TAC TA-3′) [18]. PCR was conducted as described by Zhao et al. [19], and the PCR conditions were 30 s at 94 °C, 30 s at 54 °C and 2 min at 72 °C for 40 cycles. The PCR product was sequenced by Genewiz (Tianjin, China). The newly obtained SSU rDNA sequences (accession number: *D. magnus* sp. nov., OR593570; *M. robustus* sp. nov., OR602728) and D2–D3 fragment of LSU rDNA sequences (accession number: *D. magnus* sp. nov., OR605587; *M. robustus* sp. nov., OR605586) have been deposited in GenBank.

SSU rDNA sequences longer than 600 bp from the family Linhomoeidae and Xyalidae retrieved from GenBank, were used in phylogenetic analysis. The final alignment consisted of 49 sequences from 5 genera in Linhomoeidae and 38 sequences from 8 genera in Xyalidae. The sequences were aligned using the Clustal W algorithm. Substitution models, specifically GTR (general time–reversible) + G (gamma distribution) for Linhomoeidae and T92 (Tamura 3–parameter) + G (gamma distribution) for Xyalidae were selected as the best–fit models. The analysis was rooted with *Bolbolaimus major* Guo, Liang, Lv, Wang, 2023 (accession number OQ835637.1). The D2–D3 fragment of LSU rDNA sequences from the families Linhomoeidae and Xyalidae, retrieved from GenBank was used. The final alignment consisted of 7 sequences from 4 genera in Linhomoeidae and 18 sequences from 8 genera in Xyalidae. Sequences were aligned using the Clustal W algorithm, and substitution model TN93 (Tamura–Nei) + G (gamma distribution) was selected as the best–fit model. The analysis was rooted with *B. major* (accession number OQ835636.1).

The maximum–likelihood (ML) analyses were performed using Mega X with 1000 bootstrap replicates. The Bayesin inference (BI) analyses were constructed with CIPRES http://www.phylo.org (accessed on 6 October 2023) and MrBayes on XSEDE v. 3.2.7a was used. The trees were run with a chain length of 10,000,000 and a burn–in fraction of 0.25. The topology of the resulting trees was visualized using FigTree v. 1.4.3 and refined with PowerPoint 2021.

## 3. Results and Discussion

*3.1. Taxonomy of Desmolaimus magnus sp. nov.*

Order Monhysterida Filipjev, 1929
Family Linhomoeidae Filipjev, 1922
Genus *Desmolaimus* de Man, 1880
*Desmolaimus magnus* sp. nov.
(Figures 1 and 2, Table 1)

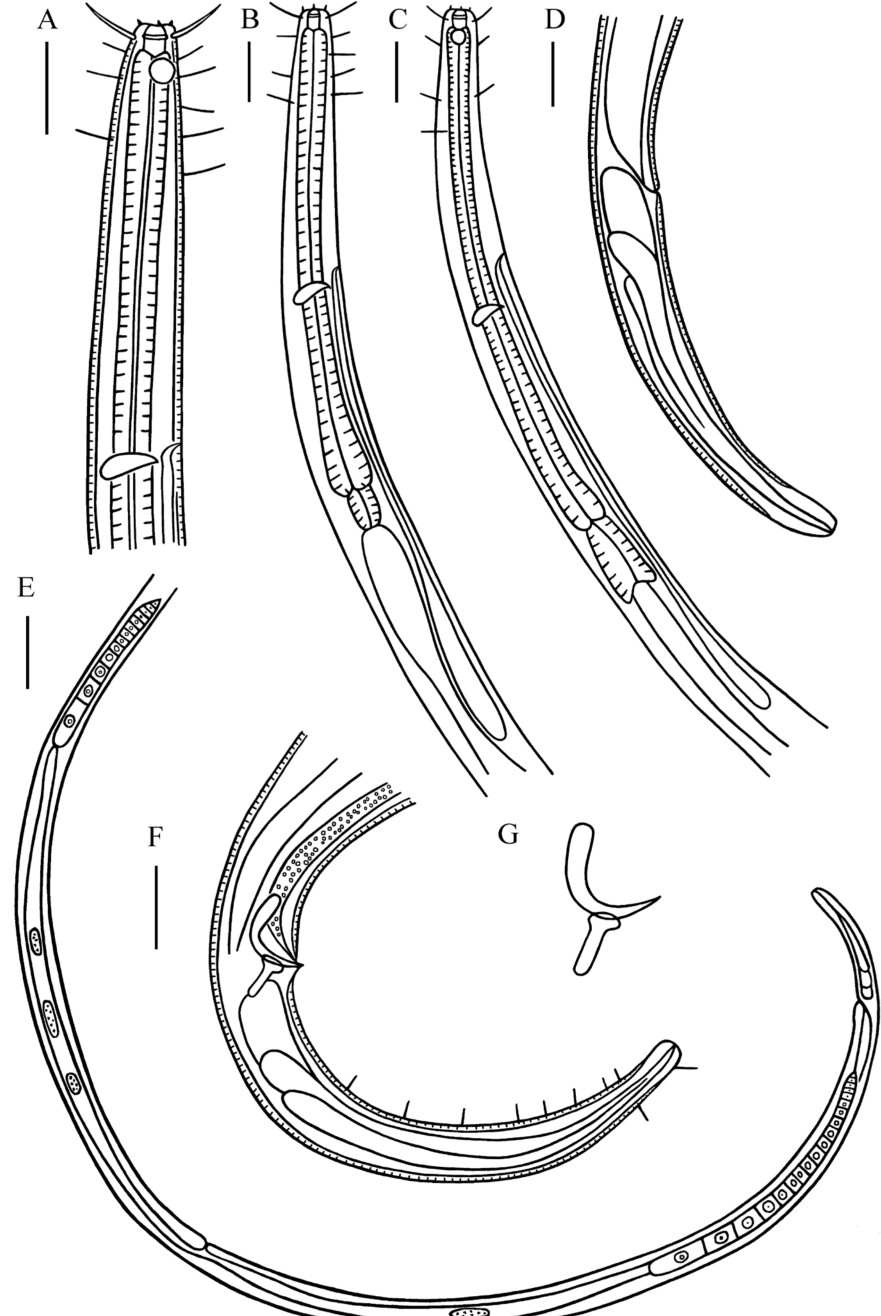

**Figure 1.** *Desmolaimus magnus* sp. nov. (**A**) lateral view of male anterior region showing buccal cavity, cephalic setae and amphidial fovea; (**B**) lateral view of male anterior region showing pharyngeal region; (**C**) lateral view of female anterior region showing buccal cavity, amphidial fovea and pharyngeal region; (**D**) lateral view of female tail; (**E**) lateral view of female posterior body showing vulva; (**F**) lateral view of male posterior body, showing spicules, gubernaculum, and tail; (**G**) spicules and gubernaculum. Scale bars: (**A**–**D**,**F**) = 30 μm; (**E**) = 100 μm.

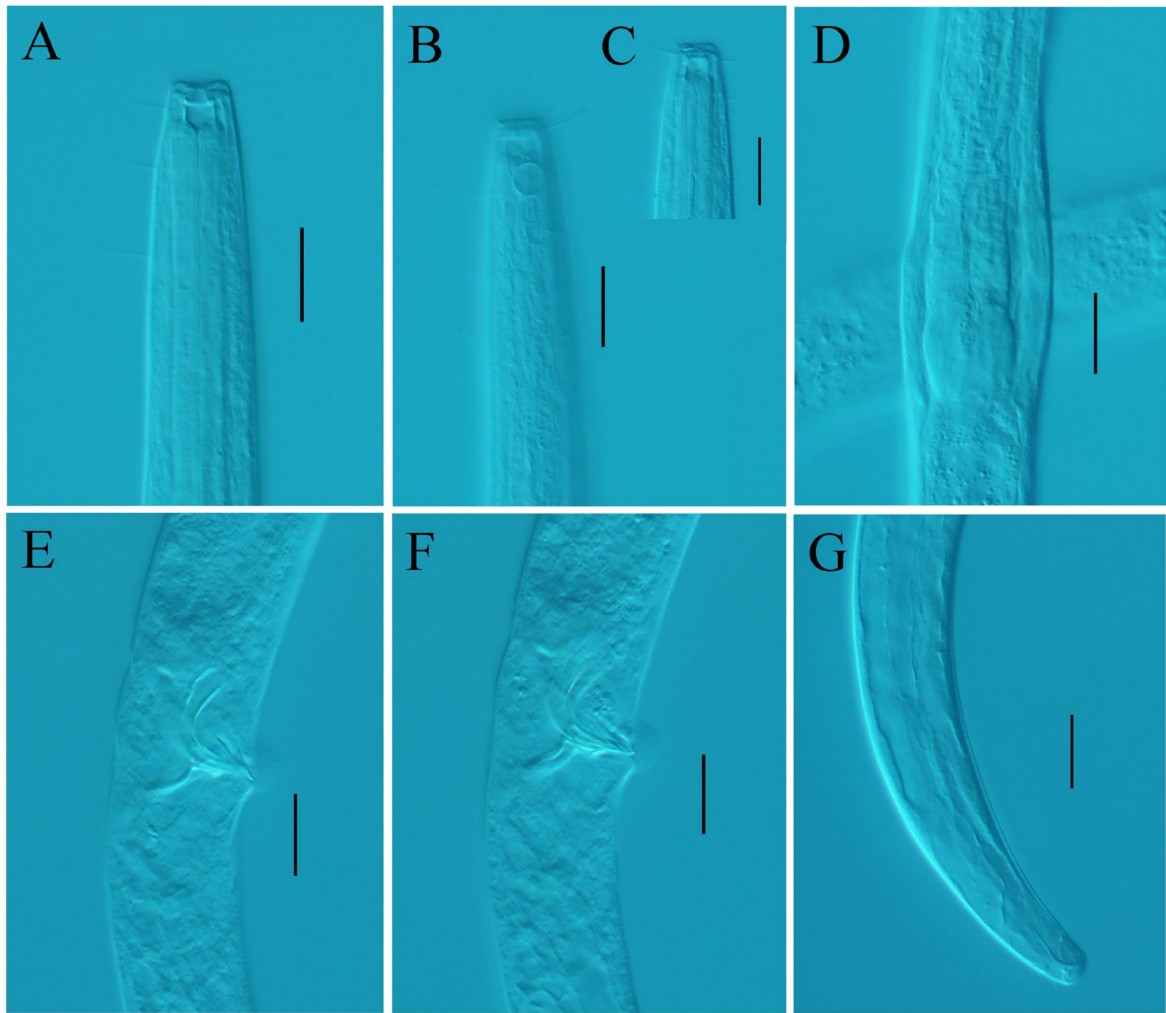

**Figure 2.** *Desmolaimus magnus* sp. nov. (**A**) lateral view of male anterior region showing buccal cavity; (**B**) lateral view of male anterior region showing amphidial fovea; (**C**) lateral view of male anterior region showing cephalic setae; (**D**) lateral view of male anterior region showing pharyngeal region; (**E**) lateral view of male posterior region showing spicules; (**F**) lateral view of male posterior region showing gubernaculum; (**G**) lateral view of male posterior region showing tail. Scale bars: (**A**–**G**) = 20 μm.

**Table 1.** Measurements of *Desmolaimus magnus* sp. nov. (in μm except for ratios).

| Characters | Holotype Male | Paratypes Males ($n = 4$) | Paratypes Females ($n = 4$) |
|---|---|---|---|
| Total body length | 4724 | 5373 ± 174 (5116–5500) | 5521 ± 357 (5008–5833) |
| Maximum body diameter | 36 | 38 ± 2 (35–40) | 43 ± 4 (40–49) |
| Head diameter | 15 | 15 ± 1 (14–15) | 15 ± 1 (14–15) |
| Length of outer labial sensilla | 3 | 3 (3–3) | 3 (3–3) |
| Length of cephalic setae | 13 | 13 (13–13) | 12 ± 1 (12–13) |
| Length of subcephalic setae | 11 | 12 ± 1 (11–12) | 12 ± 2 (11–14) |
| Amphidial fovea width | 8 | 8 (8–8) | 8 (8–8) |
| Amphidial fovea from anterior end | 11 | 10 ± 1 (9–11) | 10 ± 1 (9–10) |
| Body diameter at amphidial fovea | 18 | 18 ± 1 (17–18) | 18 (18–18) |
| Nerve ring from anterior end | 147 | 163 ± 10 (150–172) | 160 ± 13 (151–178) |
| Body diameter at nerve ring | 31 | 33 ± 1 (31–34) | 31 ± 2 (29–34) |
| Pharynx length | 266 | 293 ± 13 (277–304) | 289 ± 14 (278–310) |
| Body diameter at base of pharynx | 36 | 36 ± 1 (35–37) | 32 ± 2 (30–35) |
| Anal body diameter | 33 | 37 ± 1 (35–38) | 30 ± 3 (27–33) |

**Table 1.** *Cont.*

| Characters | Holotype Male | Paratypes Males (*n* = 4) | Paratypes Females (*n* = 4) |
|---|---|---|---|
| Spicules length along arc | 40 | 40 ± 1 (39–41) | - |
| Gubernaculum length | 18 | 18 ± 2 (16–20) | - |
| Vulva from anterior end | - | - | 3243 ± 251 (2910–3473) |
| Body diameter at vulva | - | - | 43 ± 4 (40–49) |
| V%, de Man's ratio | - | - | 58.7 ± 1.4 (57.1–60.1) |
| Tail length | 225 | 250 ± 7 (241–256) | 232 ± 8 (223–239) |
| a, de Man's ratio | 131.2 | 140.9 ± 10.8 (134.6–157.1) | 129.1 ± 13.1 (114.2–141.2) |
| b, de Man's ratio | 17.8 | 18.4 ± 1.2 (16.9–19.6) | 19.1 ± 1.4 (17.8–20.3) |
| c, de Man's ratio | 21.0 | 21.5 ± 1.1 (20.1–22.6) | 23.8 ± 1.3 (22.5–25.6) |
| c', de Man's ratio | 6.8 | 6.8 ± 0.4 (6.3–7.3) | 7.7 ± 0.6 (7.1–8.3) |

LSIDurn:lsid:zoobank.org:act:D5D54F33-2679-49D5-A3A4-F38E499BA3ED

Type material: Five males and four females were measured and studied. Holotype: m#1 on slide 22LJW3–2–4; paratypes: m#2, m#3 on 22LJW3–2–4, m#4, m#5 on 22LJW3–2–8; f#1, f#2, f#3 on 22LJW3–2–3 and f#4 on 22LJW3–1–9.

Type locality and habitat: Rizhao coast, Shandong Province, China. 35°18′ N, 119°31′ E (LJW), 0–2 cm sediment depth, sandy sediment.

Etymology: The term "magnus" refers to the large body size.

Measurements: All measurement data can be found in Table 1.

### 3.1.1. Description

Males. Body cylindrical and relatively large, anterior, and posterior body ends narrowed. Cuticle finely striated. Six inner labial sensilla papilliform and six outer labial sensilla setiform, 3 μm in length (0.20–0.21 head diameter) in two separate circles. Four cephalic setae, 0.87–0.93 head diameter in length, situated at the level of cuticularized ring. Four subcephalic setae, 11 μm in length, situated at the level of amphidial fovea, somatic setae only present in anterior pharynx (11–12 μm in length) and at tail region (5 μm in length). Head blunt. Buccal cavity relatively large, 10–12 μm in depth with cuticularized walls. Pharyngostoma relatively extensive, with cuticularized transverse rings. Amphidial fovea circular (44.4–47.1% corresponding body diameter–c.b.d.), situated at junction of buccal cavity and pharynx, 0.60–0.79 head diameter from anterior end. Pharynx cylindrical, posterior region muscular and swollen but without forming basal bulb. Nerve ring slightly posterior to middle pharynx region (54.2–56.3% of pharynx length). Cardia muscular, 22–23 μm in length. Secretory–excretory system present, renette cell situated slightly posterior to cardia, excretory pore located posterior to nerve ring.

Reproductive system diorchic, testes opposed and outstretched. Spicules paired and strongly curved, 1.05–1.21 cloacal body diameters along arc, with slightly swollen proximal end and pointed distal end. Gubernaculum short with dorso–caudally directed apophysis. Precloacal supplements not observed. Tail elongated conical with a truncated tip, 6.34–7.31 cloacal body diameters in length. Tail with two longitudinal rows of subventral setae, 6–9 per row, 1–2 short caudal setae on dorsal side and two setae at terminal end (6 μm in length). Three caudal glands present.

Females. Similar to males in most characteristics. Reproductive system didelphic, with opposed and outstretched ovaries. Anterior ovary to the right and posterior ovary to the left of intestine. Vulva slightly at the posterior of the total body. Vagina short and surrounded by constrictor muscle. Tail elongated conical without caudal setae.

### 3.1.2. Differential Diagnosis and Discussion

*Desmolaimus magnus* sp. nov. is characterized by its large body size, finely striated cuticle, a large buccal cavity with cuticularized transverse rings, six outer labial sensilla setiform, and four long cephalic setae. It also features an amphidial fovea located at

junction of the buccal cavity and pharynx. The pharynx lacks a bulb, and the spicules are curved with a slightly swollen proximal end. Additionally, there is a gubernaculum with a dorso–caudal apophysis, and the tail is elongated and conical with short caudal setae.

*Desmolaimus magnus* sp. nov. differs from other species of the genus *Desmolaimus* in body length, the location of the amphidial fovea at the junction of the buccal cavity and pharynx, and the absence of a pharyngeal bulb. *D. magnus* sp. nov. is similar to *D. brasiliensis*, *D. courti*, and *D. zosterae* in spicules, gubernaculum, and tail shape. But it differs from *D. brasiliensis* in amphidial fovea diameter (44.4–47.1% c.b.d. vs. 35% c.b.d.), spicules shape, and length (proximal end slightly widened, 39–41 μm vs. proximal end with ventral interruption, 20 μm), and gubernaculum apophysis shape (straight dorsal caudal apophysis vs. wave–shaped dorsal apophysis) [8]; differs from *D. courti* in subcephalic setae (4 vs. 8), higher de Man's ratio a (114.2–157.1 vs. 67.0–81.9), buccal cavity shape (one cuticularized rings vs. two cuticularized rings), amphidial fovea width (44.4–47.1% c.b.d. vs. 31–35% c.b.d.), spicules shape and length (proximal end slightly widened, 39–41 μm vs. proximal end not widened, 22–27 μm), and tail length (6.34–7.31 cloacal body diameters vs. 4.50–5.10 cloacal body diameters) [9]; differs from *D. zosterae* in higher de Man's ratio a (114.2–157.1 vs. 30.0–43.5), longer cephalic setae (11–13 μm vs. 7 μm), amphidial fovea width (44.4–47.1% c.b.d. vs. 28.5% c.b.d.), spicules shape (slightly swollen at the proximal end vs. proximal end widened in the form of button), and tail length (7.13–8.26 cloacal body diameters vs. 4.5 cloacal body diameters) [20].

### 3.1.3. Molecular Phylogenetic Analysis

The ML topology based on SSU and LSU rDNA D2–D3 fragment sequences are in accordance with the BI topology, and only the BI trees are shown in Figures 3 and 4.

Five genera, *Eleutherolaimus* Filipjev, 1922, *Desmolaimus*, *Linhomoeus* Bastian, 1865, *Metalinhomoeus* de Man, 1907 and *Terschellingia* de Man, 1888 are included in the SSU analysis and only the genus *Terschellingia* forms a well–supported monophyletic clade (100% posterior probability and 81% bootstrap value), constituting a sister clade to the other genera within Lihomoedae. This is in accordance with Leduc & Zhao [21] probably for its minute or conical shaped buccal cavity in *Terschellingia* distinguishing from cup–shaped buccal cavity of the other genera. Sequences of *Terschellingia longicaudata* were not shown as monophyletic clade in accordance with Leduc & Zhao and Sahraen for the high intraspecific variation [21,22]. *Metalinhomoeus* sp. (MK626797.1) and *Terschellingia lutosa* (MK626784.1) constitute a highly supported clade (100% posterior probability and 100% bootstrap value) and are identical to each other. These two sequences were submitted by the same author and should be retained for further examination.

Four species of *Desmolaimus* have been identified to the species level: *D. brasiliensis*, *D. longicaudatus*, *D. magnus* sp. nov., and *D. zeelandicus*. They are shown as a paraphyletic group. *D. magnus* sp. nov. forms a highly supported clade with *Eleutherolaimus paraschneideri* Leduc & Zhao, 2023 in the topology trees based on both SSU and LSU sequences (100% posterior probability and 97% bootstrap value, 100% posterior probability and 100% bootstrap value, respectively). These two species share similarities in amphidial fovea position, absence of a pharynx bulb, gubernaculum shape, and a conical tail with truncated end. However, they can be easily distinguished by the presence or absence of a cuticularized ring in the buccal cavity and the arrangement of cephalic setae.

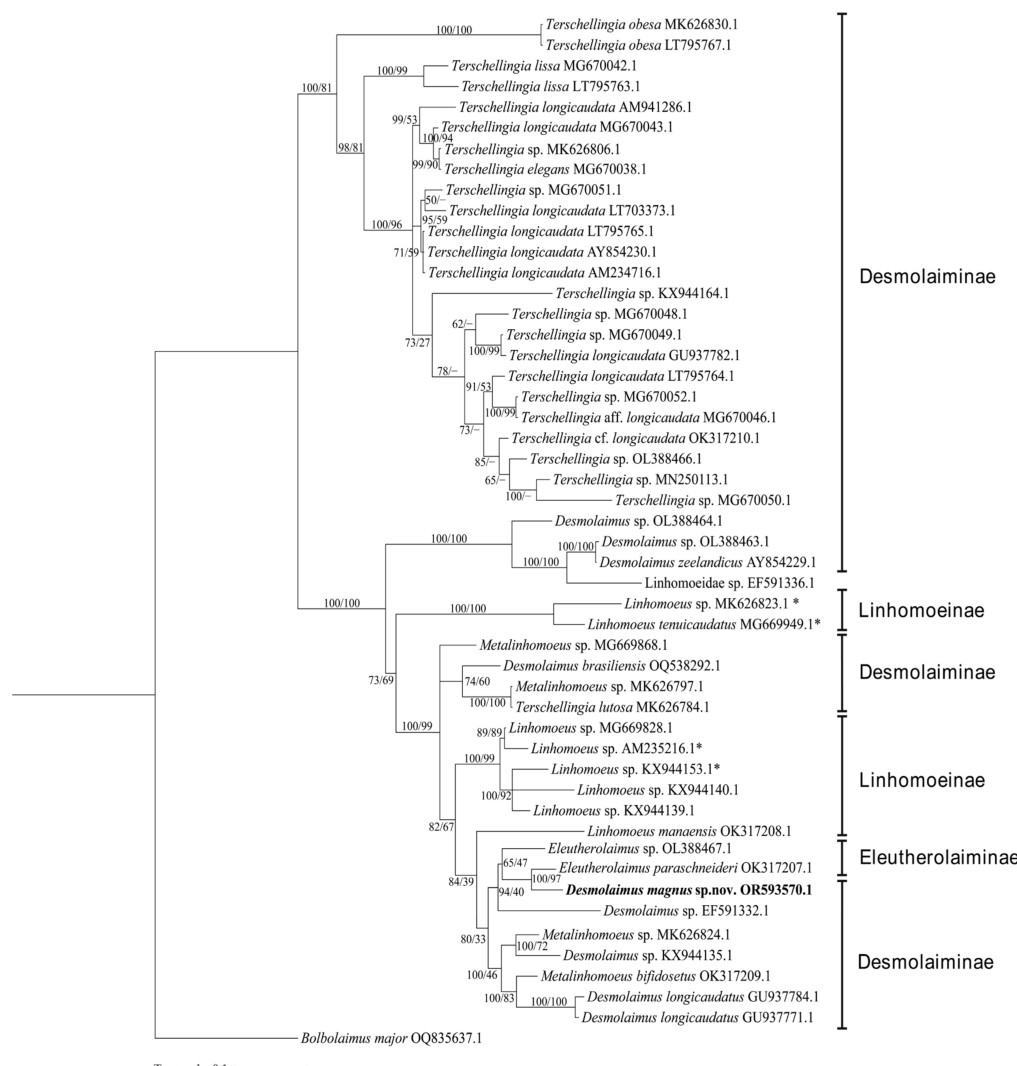

**Figure 3.** Bayesian inference tree within the family Linhomoeidae inferred from small subunit (SSU) sequences under the GTR(general time–reversible) + G (gamma distribution) model. Posterior probability on the left and bootstrap values on the right are provided for corresponding clades. The sequence obtained in this study is displayed in bold. Subfamilies are listed on the right. Note: * indicates species of the genus *Paralinhomoeus* accepted as *Linhomoeus*. The scale represents substitutions per site.

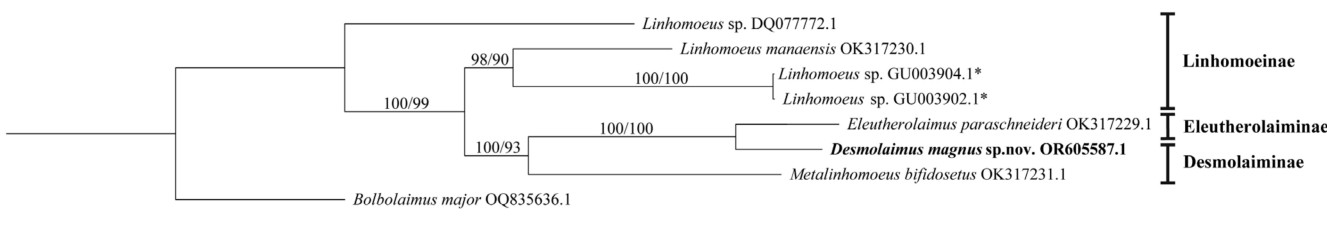

**Figure 4.** Bayesian inference tree within the family Linhomoeidae inferred from the D2–D3 fragment of large subunit (LSU) sequences under the TN93(Tamura–Nei) + G(gamma distribution) model. Posterior probability on the left and bootstrap values on the right are provided for corresponding clades. The sequence obtained in this study is displayed in bold. Subfamilies are listed on the right. Note: * indicates species of genus *Paralinhomoeus* accepted as *Linhomoeus*. The scale represents substitutions per site.

*3.2. Taxonomy of Metadesmolaimus robustus* sp. nov.

Order Monhysterida Filipjev, 1929
Family Xyalidae Chitwood, 1951
Genus *Metadesmolaimus* Schuurmans Stekhoven, 1935
*Metadesmolaimus robustus* sp. nov.
(Figures 5 and 6, Table 2)

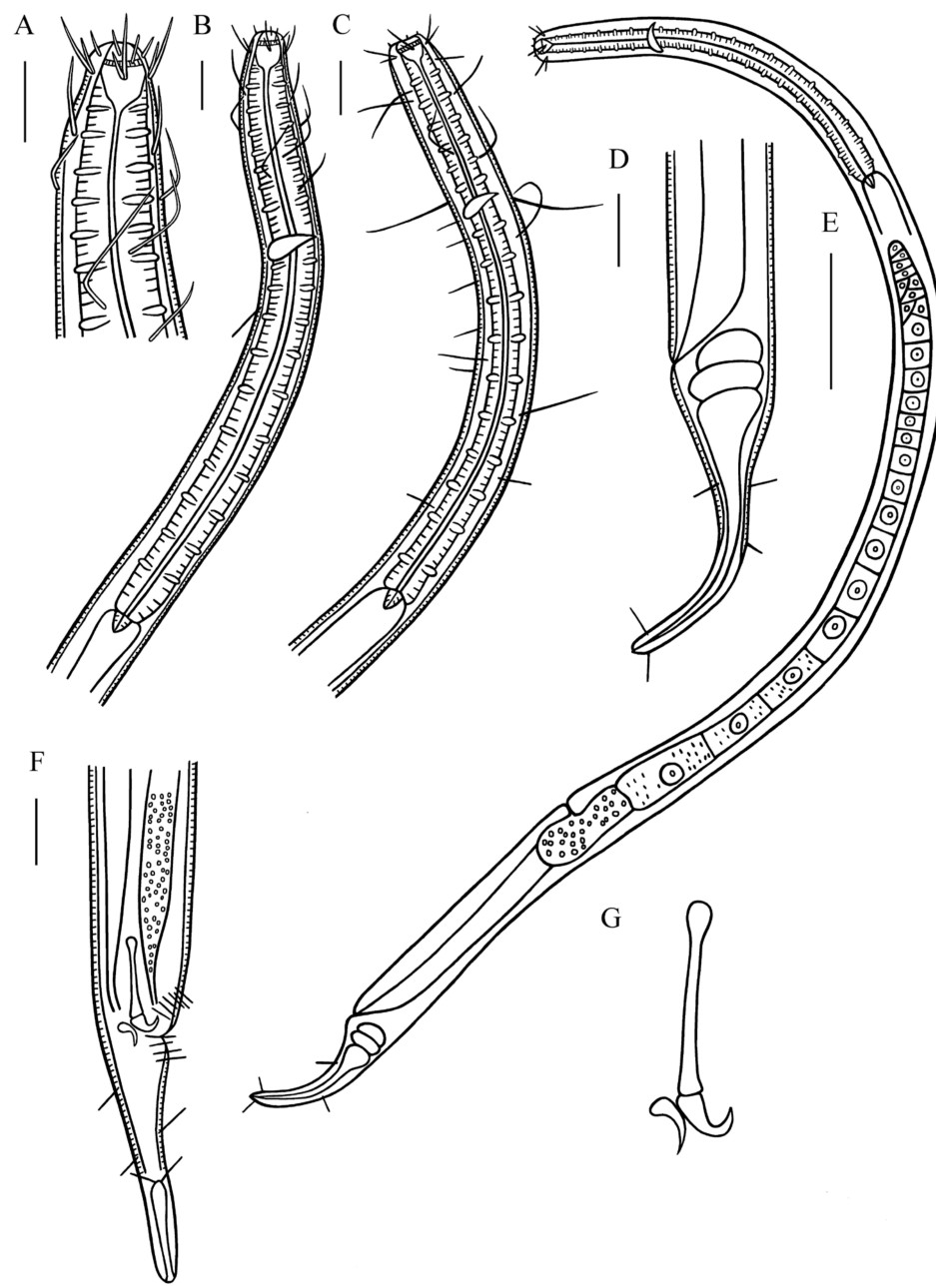

**Figure 5.** *Metadesmolaimus robustus* sp. nov. (**A**) lateral view of male anterior region showing cephalic setae and buccal cavity; (**B**) lateral view of male anterior region showing pharyngeal region; (**C**) lateral view of female anterior region showing buccal cavity and pharyngeal region; (**D**) lateral view of female posterior body, showing tail; (**E**) lateral view of female entire body showing vulva; (**F**) lateral view of male posterior body, showing spicules, gubernaculum and tail; (**G**) spicules and gubernaculum. Scale bars: (**A–D,F**) = 30 μm; (**E**) = 100 μm.

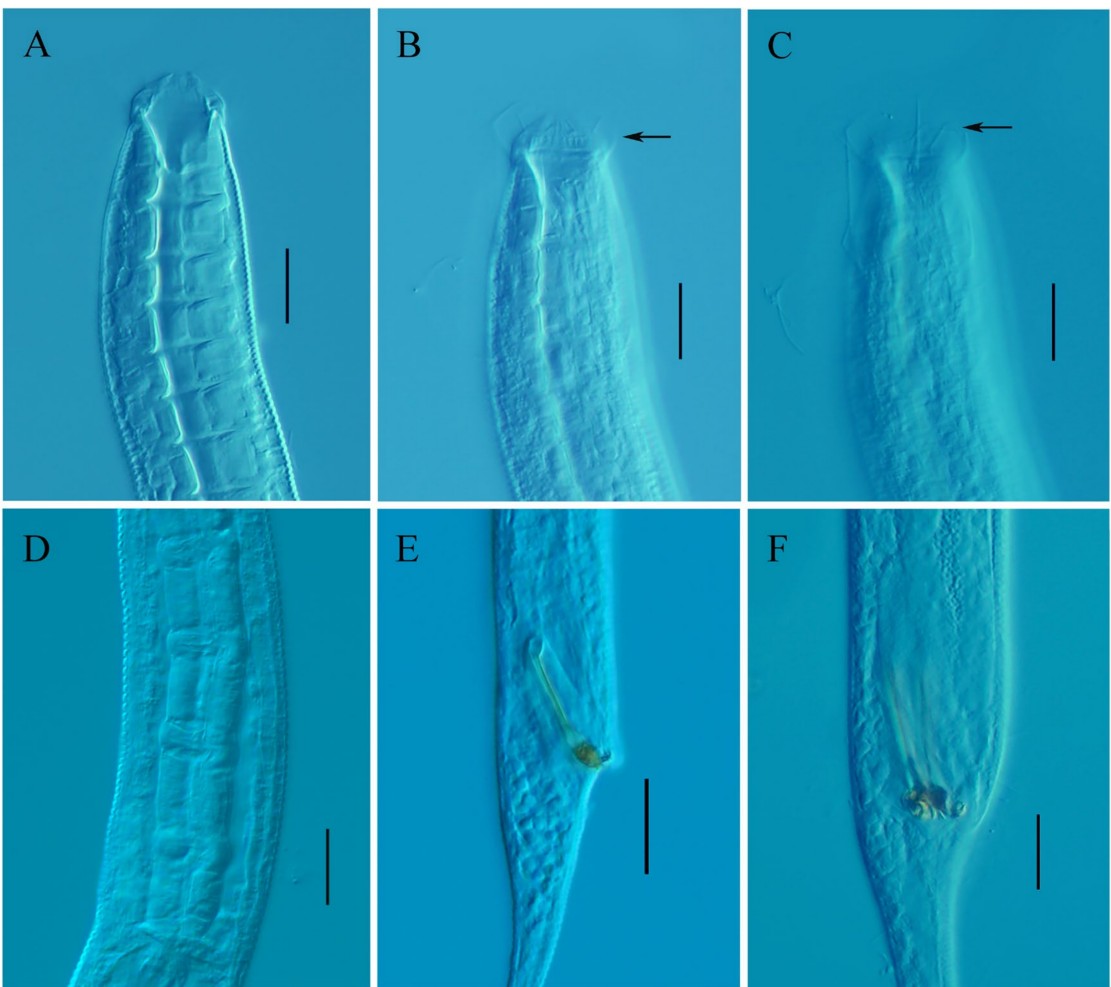

**Figure 6.** *Metadesmolaimus robustus* sp. nov. (**A**) lateral view of male anterior region showing buccal cavity; (**B**) lateral view of male anterior region showing inner labial sensilla and cuticularized rings (arrow); (**C**) lateral view of male anterior region showing ventrolateral setiform (arrow); (**D**) lateral view of female anterior body, showing pharyngeal region; (**E**) lateral view of male posterior region showing spicules; (**F**) lateral view of male posterior region, showing gubernaculum. Scale bars: (**A–F**) = 20 μm.

**Table 2.** Measurements of *Metadesmolaimus robustus* sp. nov. (in μm except for ratios).

| Characters | Holotype Male | Paratypes Males ($n = 3$) | Paratype Female ($n = 1$) |
|---|---|---|---|
| Total body length | 2084 | 1546 ± 66 (1471–1593) | 1697 |
| Maximum body diameter | 43 | 32 ± 2 (31–34) | 52 |
| Head diameter | 25 | 22 ± 1 (21–23) | 26 |
| Length of inner labial sensilla | 7 | 6 (6–6) | 7 |
| Length of outer labial sensilla | 23 | 21 ± 1 (20–21) | 21 |
| Length of cephalic setae | 15 | 11 ± 1 (11–12) | 12 |
| Nerve ring from anterior end | 142 | 89 ± 1 (88–90) | 132 |
| Body diameter at nerve ring | 42 | 28 ± 2 (27–30) | 39 |
| Pharynx length | 420 | 312 ± 11 (303–324) | 416 |
| Body diameter at base of pharynx | 43 | 31 ± 1 (30–32) | 43 |
| Anal body diameter | 35 | 26 ± 1 (26–27) | 34 |
| Spicules length along arc | 52 | 41 ± 2 (39–42) | – |
| Gubernaculum length | 12 | 10 ± 1 (9–10) | – |
| Vulva from anterior end | – | – | 1233 |

**Table 2.** *Cont.*

| Characters | Holotype Male | Paratypes Males (*n* = 3) | Paratype Female (*n* = 1) |
|---|---|---|---|
| Body diameter at vulva | – | – | 51 |
| V%, de Man's ratio | – | – | 72.66% |
| Tail length | 173 | 126 ± 7 (122–134) | 151 |
| a, de Man's ratio | 48.5 | 47.9 ± 1.2 (46.9–49.2) | 32.6 |
| b, de Man's ratio | 5.0 | 4.9 ± 0.3 (4.7–5.2) | 4.1 |
| c, de Man's ratio | 12.0 | 12.3 ± 0.5 (11.9–12.9) | 11.2 |
| c', de Man's ratio | 4.9 | 4.8 ± 0.4 (4.5–5.2) | 4.4 |

LSIDurn:lsid:zoobank.org:act:901DC71D-C907-441A-A4B7-4E146D699F27

Type material: Four males and one female were measured and studied. Holotype: m#1 on slide 22SHT5–2–11; paratypes: m#2 on 22RZDT4–2–8, m#3 on 22RZDT4–1–9, m#4 on 22SHT5–1–28; f#1 on 22RZDT4–3–6.

Type locality and habitat: Rizhao cost, Shandong Province, China. 35°26′ N, 119°34′ E (SHT); 35°23′ N, 119°33′ E (RZDT), 0–2 cm sediment depth, sandy sediment.

Etymology: The term "robustus" refers to the strongly muscularized pharynx region.

Measurements: All measurement data can be found in Table 2.

### 3.2.1. Description

Males. Body cylindrical and long. Cuticle coarsely striated in 2 μm, not conspicuously brown in color. Anterior region slightly setoff, lips high. Six inner labial sensilla setiform, 6–7 μm in length (0.26–0.29 head diameter). Six outer labial setae, 20–23 μm in length (0.91–0.95 head diameter) and four cephalic setae, 11–15 μm in length (0.48–0.60 head diameter) in a same circle. Two lateral setae present next to outer labial setae, 0.48–0.60 head diameter in length, form 6 + 12 arrangement. Ventrolateral setiform structure exists between labial and cephalic setae. Entire pharynx region with numerous somatic setae of various lengths, 15–60 μm in length. Buccal cavity relatively spacious with cuticularized walls, 22–27 μm in depth, 19 μm in width, divided into two parts by cuticularized rings. Amphidial fovea not observed. Pharynx cylindrical, strongly muscularized and divided into 16–20 parts (15–19 μm) without pharynx bulb. Nerve ring anterior to the pharynx region (27.5–33.8% of pharynx length). Cardia short. Excretory pore not observed.

Reproductive system with single outstretched anterior testis to right of intestine. Spicules paired and equal, straight with proximal end slightly swollen, distal portion widened and jointed with hooked distal end, 1.49–1.58 cloacal body diameters long. Gubernaculum brownish, comma shaped. Cloacal setae in one circle, 9–10 μm in length. Precloacal supplement not observed. Tail conico–cylindrical, 4.52–5.15 cloacal body diameters in length, conical part short (35.8–40.2% of tail length). Two terminal setae present, 15–20 μm in length. Caudal glands not observed.

Female. Similar to males in most characteristics. Reproductive system with single outstretched ovary to left of intestine. Vulva at posterior of total body. Vagina short. Three caudal glands in line.

### 3.2.2. Differential Diagnosis and Discussion

*Metadesmolaimus robustus* sp. nov. is characterized by its large body size, coarsely striated cuticle, spacious buccal cavity with cuticularized rings, six inner labial sensilla setiform, six outer labial setae, and four long cephalic setae. The pharynx is muscular and lacks a bulb. Spicules are straight with a slightly swollen proximal end and a jointed distal portion with a hooked distal end. The gubernaculum has a comma shape, and the tail is conico–cylindrical with a short conical part.

*Metadesmolaimus robustus* sp. nov. differs from other species of the genus *Metadesmolaimus* by having jointed spicules and a comma–shaped gubernaculum. *M. robustus* sp. nov. is similar to *M. communis*, *M. hamatus*, and *M. zhanggi* in having straight spicules. However,

it differs from *M. communis* in body length (1471–2084 μm vs. 827–932 μm), higher de Man's ratio a (46.9–49.2 vs. 22–28), longer inner labial sensilla (6–7 μm vs. 1.0–1.5 μm), longer outer labial sensilla (20–23 μm vs. 7–9 μm), longer spicules (39–52 μm vs. 27–29 μm), and a different tail shape (conico–cylindrical vs. elongated conical) [23]. It differs from *M. hamatus* in inner labial setae length (6–7 μm vs. papilliform), outer labial sensilla length (20–23 μm vs. 10–11 μm), spicules length (39–52 μm vs. 28 μm), and tail shape (conico–cylindrical vs. elongated conical) [24]. It also differs from *M. zhanggi* in body length (1471–2084 μm vs. 930–980 μm), the shape of inner and outer labial sensilla (unjointed vs. jointed), and spicules length (39–52 μm vs. 24–29 μm) [15].

### 3.2.3. Molecular Phylogenetic Relationships and Analysis

The ML topology based on SSU and LSU rDNA D2–D3 fragment sequences closely aligns with the BI topology. As a result, only the BI trees are presented in Figures 7 and 8. In the SSU analysis, only sequences identified to the species level are retained, with the exception of the genus *Metadesmolaimus*, which includes *M. robustus* sp. nov. This selection is made due to the high morphological similarity among genera.

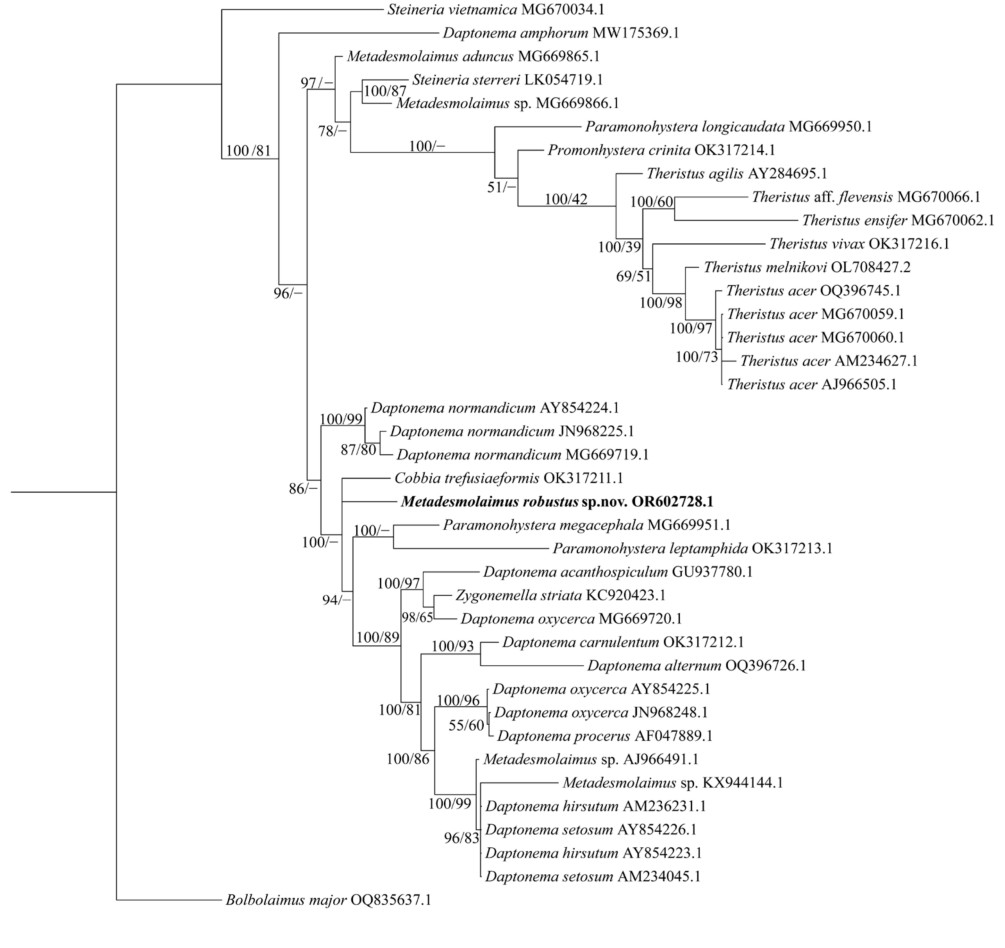

**Figure 7.** Bayesian inference tree within the family Xyalidae, inferred from small subunit (SSU) sequences using the T92 (Tamura 3–parameter) + G (gamma distribution) model. Posterior probability on the left and bootstrap values on the right are provided for the corresponding clades. The sequence obtained in this study is highlighted in bold. The scale represents substitutions per site.

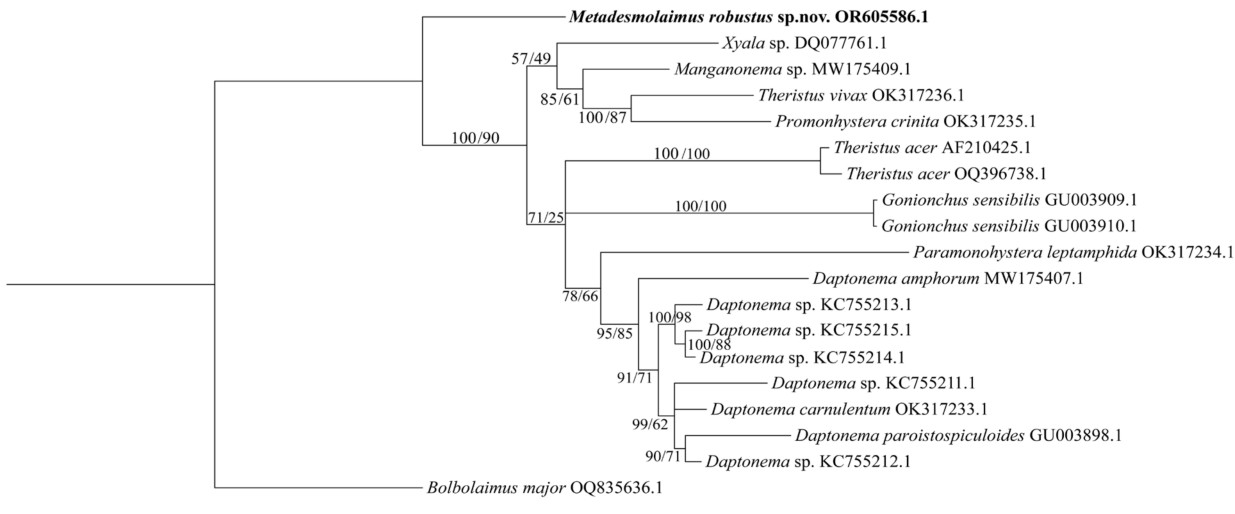

**Figure 8.** Bayesian inference tree within the family Xyalidae, inferred from D2–D3 fragment of large subunit (LSU) sequences using the TN93 (Tamura–Nei) + G (gamma distribution) model. Posterior probability on the left and bootstrap values on the right are provided for the corresponding clades. The sequence obtained in this study is highlighted in bold. The scale represents substitutions per site.

Eight genera, *Cobbia* de Man, 1907, *Daptonema*, *Metadesmolaimus*, *Paramonohystera* Steiner, 1916, *Promonhystera* Wieser, 1956, *Steineria* Micoletzky, 1922, *Theristus*, and *Zygonemella* Cobb, 1920 are included in the SSU analysis. Notably, *Steineria vietnamica* Gagarin, 2013 (MG670034.1) stands apart from the main Xyalidae clade. This distinction can be attributed to the unique cuticle shape of *S. vietnamica*, characterized by cuticle annules in resolvable dots, setting it apart from other Xyalid species [25]. In the topology tree, *S. vietnamica* does not cluster with *Steineria sterreri*, and its precise position may require further discussion, especially with the availability of more molecular sequences from the other four *Steineria* species exhibiting the same cuticle feature.

Genera *Daptonema* and *Theristus* are the most complex genera within the family Xyalidae, with minor morphological differences that primarily revolve around tail form and the terminal setae at the tail tip [4]. In the SSU analysis, ten species of *Daptonema* have been identified to the species level and are shown as paraphyletic groups. Notably, *Daptonema amphorum* Leduc, 2015 (MW175369.1) and *D. normandicum* split from the main clade of the other eight *Daptonema* species (99% posterior probability and 83% bootstrap value, 86% posterior probability, respectively) in accordance with Leduc [26]. Meanwhile, *Daptonema setosum* (Bütschli, 1874) Lorenzen, 1977 and *D. hirsutum* (Vitiello, 1967) Lorenzen, 1977 show a close relationship in SSU molecular analyses. Neres et al. considered *D. setosum* as a junior synonym of *D. hirsutum* based on molecular and morphological analyses, this is also supported in our SSU phylogenetic tree [27]. In the LSU phylogenetic tree, *Daptonema* sequences also form a well–supported clade (95% posterior probability and 85% bootstrap value). In the SSU topology tree, six species of genus *Theristus* form a moderately supported monophyletic group (100% posterior probability and 42% bootstrap value), although this is not supported in the LSU tree. The SSU topology tree for *Daptonema* and *Theristus* largely aligns with Leduc [26], but it depicts *Theristus* as a monophylectic clade within the family Xyalidae. Conversely, the LSU topology tree indicates that the genus *Daptonema* forms a monophylectic group within the family Xyalidae.

In the SSU analysis, five sequences of the genus *Metadesmolaimus* are included with only *M. aduncus* and *M. robustus* sp. nov. identified to the species level. These two species are depicted as paraphyletic clades. In the SSU topology tree, *M. robustus* sp. nov. is also shown as a sister clade to genera *Daptonema*, *Zygonemella*, *Cobbia* and *Paramonohystera*. In

the LSU topology tree, *M. robustus* sp. nov. diverges early from the main Xyalidae clade, confirming its status as a new species.

The genus *Zygonemella* exhibits morphological similarities to the genus *Daptonema*, with *Z. striata* Cobb, 1920 being the sole species within *Zygonemella*. *Z. striata* clusters with *Daptonema oxycerca*, forming a moderately supported clade (98% posterior probability and 65% bootstrap value). This lends support to the conclusion that *Z. striata* is synonymous with the genus *Daptonema* [28].

## 4. Conclusions

*Desmolaimus magnus* sp. nov. can be distinguished from other species in the genus *Desmolaimus* by its larger body size, possession of six outer labial sensilla setiform, and four long cephalic setae, an amphidial fovea located at the junction of the buccal cavity and pharynx, a pharynx without a bulb, curved spicules with a slightly swollen proximal end, a gubernaculum with a dorso–caudal apophysis, and an elongated conical tail with short caudal setae. Phylogenetic analysis confirms *D. magnus* sp. nov. as a new species, while it also reveals that the genus *Desmolaimus* forms a paraphyletic group and the genus *Terschellingia* constitutes a monophyletic group. With the description of *D. magnus* sp. nov., a total of nine species of *Desmolaimus* have been identified.

*Metadesmolaimus robustus* sp. nov. sets itself apart from other species in the genus *Metadesmolaimus* with its large body size, possession of six inner labial sensilla setiform, six long outer labial setae, four long cephalic setae, a muscularized pharynx without a bulb, straight and jointed spicules, brownish, and a comma–shaped gubernaculum. Phylogenetic analysis confirms the status of *M. robustus* sp. nov. as a new species. It also reveals that the genus *Metadesmolaimus* forms a paraphyletic group, while the genus *Theristus* is a monophyletic group based on SSU sequence data, and the genus *Daptonema* is a monophyletic group based on LSU sequence data. Furthermore, the analysis supports the conclusion that *Z. striata* is synonymous with the genus *Daptonema*. With the description of *M. robustus* sp. nov., a total of 16 species of *Metadesmolaimus* have been identified.

**Author Contributions:** Species description and molecular phylogenetic analysis, W.G.; slides making and picture processing, Z.M.; manuscript preparation, C.W. All authors have read and agreed to the published version of the manuscript.

**Funding:** This work was supported by A Project of Shandong Province Higher Educational Science and Technology Program (J18KA152) and Open Project of Liaocheng University Animal Husbandry Discipline (319312101).

**Institutional Review Board Statement:** Not applicable.

**Data Availability Statement:** All data are available in the article.

**Acknowledgments:** We are greatly thankful to three anonymous reviewers for their kind reviews and valuable suggestions and Yanwei Lv in sample collection.

**Conflicts of Interest:** The authors declare no conflict of interest.

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
