# Peer review of "Two New Nematode Species, Desmolaimus magnus sp. nov. (Monhysterida, Linhomoeidae) and Metadesmolaimus robustus sp. nov. (Monhysterida, Xyalidae), from the Yellow Sea, China with Phylogenetic Analyses within Linhomoeidae and Xyalidaeâ€"

_diversity, doi:10.3390/d15111130_

Round 1
Reviewer 1 Report
Comments and Suggestions for Authors
General opinion
The manuscript describes two new species of marine nematodes including the assessment of the phylogenetic position within their respective families. The manuscript is well-written and follows the standard practice of the modern taxonomy of nematodes. Authors provide solid evidence of the discovery of the new species and the conclusions are supported by the gathered data. This is valuable research of interest for the international community of taxonomists and is a relevant contribution to the discovery of nematode diversity. The main shortcoming of the study is that the discussion of phylogenetic relationships within families is too short. Authors should expand a bit and discuss the implication of the tree topologies in the current classification system of Nematoda. For instance, they may suggest potential changes in current classification, and misplaced genera.
I provide several remarks below that may improve the manuscript.
Specific remarks
L21. Abstract. “…confirm the taxonomic position…” Confirmation is not the best word since this are new species. I recommend changing with something like: … sequences placed the new species within the phylogenetic tree (or framework) of the families…
L37-41. This paragraph fits better at the end of the introduction as objectives/aims. Otherwise, objectives should be stated explicitly at the end of introduction since this section ends abruptly.
L91. Please, provide information about the sampling, for instance, type of habitat, environmental conditions, depth, and sampling device.
L99-101.I would like to see more details about the taxonomic practice. For instance, how measurements were done, what software was used for drawings, camera model, etc.
L115. Please, provide full details about PCR conditions.
L133. Spell in full ML and BI.
L149. Please, inform about the microhabitat of the species (sediment or hard substrate).
L150.What is LJW?
L153. Table 1. The precision of calculations should not be higher than the source data. For instance, 5372.5 μm is giving an unreal precision, it should be 5373 μm. Some well-known abbreviations (e.g., V%, De Man’s ratios) should be explained in full.
L174. 6-9 per row.
L193. For clarity, use De Man’s ratio a instead of “a”.
L221 Fix typo: Tershellingia.
L228. Fix typo: paraphylectic,
L231. Add “bootstrap value, respectively…”.
L240. Caption figure 3. Fix typos: Paralinhomeous, Linhomeous. Also, in caption of figure 4.
L257. What are SHT and RZDT?
L271. Remove “a”.
L308-309. Fix typos in the sentence.
L321. A paraphyletic clade does not exist, clade is a monophyletic group by definition. The same at L347, L373
L342. Typo: Thersitus.
L356. Figure 7. M. robustus sp. nov is not in bold.
L373. Typo: paraphylectic. Also at L380.

No comments
Author Response
- As to the suggestion to Line 21, in the abstract, we adjusted the sentence according to the reviewer’s advice in line 22-23.
- As to the suggestion to line 37-41, we moved the part to line Line 90-94 as “During our study of marine nematode diversity in Rizhao coast, the Yellow Sea, China, two new species, Desmolaimus magnus nov. and Metadesmolaimus robustus sp. nov. are described and sequences of SSU and D2–D3 fragment of LSU rDNA are acquired, phylogenetic analysis among genera within family Linhomoeidae and Xyalidae are analyzed to clarify genus position.” is adjusted here according to the reviewer’s advice” according to the reviewer’s advice.
- As to the suggestion to line 91, “using a syringe with a 2.6 cm inner diameter to a depth of 10 cm ” was added to give the sampling depth, and sampling device.
- As to the suggestion to line 99-101, “with camera of Axiocam 208 color, line drawings were made with Labscope software through an iPad (Apple, USA), and measurements and”was added according to the reviewers’ advice to give more details about the taxonomic practice in line 108-110.
- As to the suggestion to line115, “The PCR conditions were 30s at 94℃, 30s at 54℃ and 2 min at 72℃ for 40 cycles.” is added according to the reviewers’ advice to provide full details about PCR conditions in line 125-126.
- As to the suggestion to line 133, “ML” is changed to “maximum–likelihood (ML)” and“BI” is changed to “Bayesin inference (BI)” according to the reviewer’s advice in line 145-146.
- As to the suggestion to line 149, “ 0-2 cm sediment depth,sandy sediment ” was added in line 161.
- As to the suggestion to line 150, “LJW” is an abbreviation for sampling station.
- As to the suggestion to Table1, we changed the number and abbreviations according to the reviewer’s advice.
- As to the suggestion to line 174, “6–9 per row” in line 186 according to the reviewer’s advice.
- As to the suggestion to line 193, “de Man’s ratio” was added in line 207.
- As to the suggestion to line 221, “Terschellingia” was corrected according to the reviewer’s advice in line 234.
- As to the suggestion to line 228, “paraphyletic” was corrected according to the reviewer’s advice in line 248.
- As to the suggestion to line 231, “bootstrap value, respectively…”is added according to the reviewers’ advice in line 251.
- As to the suggestion to line 240, “Paralinhomoeus” and “Linhomeous” are corrected according to the reviewers’ advice in line 262 and 268.
- As to the suggestion to line 257, “SHT and RZDT” are abbreviation for sampling station.
- As to the suggestion to line 271, “a” is removed according to the reviewers’ advice.
- As to the suggestion to line 208-309, “Cobbia” is adjusted according to the reviewers’ advice in line 333.
- As to the suggestion to line 321, we changed the expression in line 346-347, and line 379; As to line373, four clades of Desmolaimus identified to species level are shown in paraphyletic clade and we keep our expression mode in 405.
- As to the suggestion to line 342, “Thersitus.” adjusted according to the reviewers’ advice in line 373.
- As to the suggestion to line 356, “ M. robustus sp. nov” is corrected according to the reviewers’ advice.
- As to the suggestion to line 373, 380, “paraphyletic” was corrected according to the reviewer’s advice in line 405, 413, respectively.
- As to the suggestion of discussion portion, we revised the portion in line 235-241, 337-341, 343-350, 367-370. and added two new references and listed in the reference list.

Reviewer 2 Report
Comments and Suggestions for Authors
Please considered improve figures of both descriptions

Author Response
- Sugestions of line 30, we changed “tropic group.” to “trophic guilds” in line 32.
- Sugestions of line 167, “Cardia muscular, relatively long (22–23 μm in length)” is changed to “Cardia muscular, 22-23 μm length.” is changed according to the reviewer’s advice in line 178.

Reviewer 3 Report
Comments and Suggestions for Authors
Specific revisions have been directly marked on the manuscript of the paper. Please refer to them.

I believe that the English used in the manuscript is well written throughout. However, despite that, I have provided specific suggestions directly in the manuscript regarding some revision points. Please refer to them.
Author Response
We accept all the comments of reviewer3, and some different changes are listed.
- As to the suggestion to line 30, “tropic group” was changed to “trophic guilds”according to the second reviewer’s advice in line 32.
- As to the suggestion to line 46, “amphids shape” was changed to “amphidial fovea shape”amphidial fovea in Line 46
- As to line 193, 198, 294 “ values of a”, we used the “de Man’s ratio a” following the second reviewer’s advice.
